# NOISE$^+$2NOISE: CO-TAUGHT DENOISING AUTOENCODERS FOR TIME-SERIES DATA

## ABSTRACT

We consider the task of learning to recover clean signals given only access to noisy data. Recent work in computer vision has addressed this problem in the context of images using denoising autoencoders (DAEs). However, to date DAEs for learning from noisy data have not been explored in the context of time-series data. DAEs for denoising images often rely on assumptions unlikely to hold in the context of time-series, *e.g.*, multiple noisy samples of the same example. Here, we adapt DAEs to cleaning time-series data with noisy samples only. To recover the clean target signal when only given access to noisy target data, we leverage a noise-free auxiliary time-series signal that is related to the target signal. In addition to leveraging the relationship between the target signal and auxiliary signal, we iteratively filter and learn from clean samples using an approach based on co-teaching. Applied to the task of recovering carbohydrate values for blood glucose management, our approach reduces noise (MSE) in patient-reported carbohydrates from $72g^2$ (95% CI: 54,93) to $18g^2$ (13,25), outperforming the best baseline (MSE $= 33g^2$ (27,43)). We demonstrate strong time-series denoising performance, extending the applicability of DAEs to a previously under-explored setting.

## 1 INTRODUCTION

**Background.** Denoising autoencoders (DAEs) (Vincent et al., 2008) have been used to accurately denoise various signals, including medical images (Gondara, 2016), ECG signals (Xiong et al., 2016), and power system measurements (Lin et al., 2019). With respect to time-series data, DAEs have been used for forecasting (Romeu et al., 2015), classification (Zheng et al., 2022) and imputation (Zhang & Yin, 2019), but generally require access to clean samples at training and do not provide de-noised outputs. In many real-world settings, clean samples are unavailable at training. Work in computer vision has addressed this problem through extensions that either require paired samples (Lehtinen et al., 2018) or rely on patch-based analysis (Krull et al., 2018; Laine et al., 2019; Xie et al., 2020; Batson & Royer, 2019). Similar approaches do not extend to time-series data, where paired samples rarely exist and patch-based techniques do not apply. Beyond approaches that rely on paired samples or patch-based analyses, researchers have recently proposed techniques that utilize knowledge of the noise distribution to recover the clean signal. These approaches either use the properties of the distribution to recover the clean signal after training on noisy data (Kim & Ye, 2021; Moran et al., 2019), or rely on the noise having low expectation and variance compared to the signal, in which case a model trained on noisy data can approximate one trained on clean data (Xu et al., 2020). While these approaches may be considered in a time-series setting (and are treated as baselines here), their applicability is limited as noise in time-series settings is rarely weak or known.

**Our Contribution.** In light of this gap, we adapt denoising autoencoders for time-series data. Our approach, 'Noise$^+$2Noise', learns to map a noisy target signal to a clean signal given only noisy samples and an auxiliary clean signal. Inspired by work in image denoising (Lehtinen et al., 2018; Xu et al., 2020), we add additional noise to the noisy target signal during training and attempt to recover the original noisy signal. Provided that the noise has low expectation and variance, a network trained in this manner can learn to recover the true signal because the noise will minimally impact the expected value of the output of the network (Xu et al., 2020). The auxiliary signal is input along with the target signal into a denoising autoencoder, which allows our network to leverage the relationship between the auxiliary and target signals. To address the fact that the signal to noise ratio might not be weak, we adapt a co-teaching approach to train two DAEs (Jiang et al., 2018;

Han et al., 2018). We use this approach to identify the cleaner samples; the most likely low-noise samples are identified as the low-loss samples of the other model and used for backpropagation. This co-teaching approach has never been applied to de-noising in a time-series or any other setting. It has also never been utilized in a continuous output setting. By adapting this co-teaching approach to DAEs, we provide a solution to denoising in this novel (time-series) setting.

**Real-world Inspiration.** Disparate levels of noise across variables are common in data streams. Measurement reliability can vary across sensors, from essentially noiseless to highly corrupted. Throughout this work we take inspiration from a real-world problem affecting millions in the US: blood glucose management. Individuals with diabetes monitor several variables over time, including their blood glucose, insulin administrations, and carbohydrate intake. Blood glucose when measured by a continuous glucose monitor (CGM) has little noise, while the number of grams of carbohydrates in a meal are patient-reported, and, as a result, are often inaccurate. Recognizing the variation in the level of noise across signals in a data stream, we propose an approach that utilizes the more reliable variables (*e.g.*, blood glucose) to update noisy variables like carbohydrates. Our approach aims to retrospectively correct measurements of patient-reported signals, like carbohydrates in the context of diabetes, which could lead to improved treatment decisions: patients can learn when they are over- or under-reporting and adjust in the future, ultimately improving health outcomes.

## 2 RELATED WORK

In denoising autoencoder training, a model input is corrupted and a network is tasked with recovering the original input. In this way, the network cannot learn the identity, unlike in basic autoencoder training (Vincent et al., 2008). Recent work in machine learning has focused on using DAEs to recover clean signals from only noisy signals. The vast majority of this work lies in image analysis and builds off of "noise2noise" (Lehtinen et al., 2018), an approach that uses multiple noisy instances of the same image to train a model to learn to denoise the image. Briefly, the approach relies on the fact that if the noise is mean zero, using a secondary noisy instance (besides the input image) as a target will produce a network that learns the clean image, in expectation, when enough training data is available. When paired samples are unavailable, others have proposed approaches based on exploiting patches sampled from the image (Krull et al., 2018; Laine et al., 2019; Xie et al., 2020; Batson & Royer, 2019), but these are not applicable to time-series data. Other approaches eschew relying on inter-variable relationships but rely heavily on a known noise function (Moran et al., 2019; Kim & Ye, 2021) or a low expectation and variance noise function (Xu et al., 2020). Our approach builds off of Xu et al. (2020), learning to reconstruct a noisy signal from only noisy samples of that signal, but in contrast to Moran et al. (2019) or Kim & Ye (2021), we do not make strong assumptions about the noise distribution. To get around these assumptions, we leverage an auxiliary signal and iteratively identify the cleaner samples within the dataset. Combined with the auxiliary signal these samples are used to train a DAE that treats these samples as ground truth.

Our approach to denoising a noisy target signal by iteratively identifying clean samples is, in part, related to work in noisy-label learning. A common approach to learning from noisy labels involves identifying and reweighting samples with clean labels during training. Samples are filtered based on gradient values (Ren et al., 2020), Jacobian ranking(Mirzasoleiman et al., 2020) or some latent state (Lee et al., 2019; Wu et al., 2020). Co-teaching (Han et al., 2018), which builds off of mentor net (Jiang et al., 2018), is performed by training two networks in parallel. Each network is back-propgated using only samples within the current mini-batch for which the loss of the *other* network is lowest. Intuitively, samples with incorrect labels are likely to have higher loss and therefore be removed. Using two networks in parallel provides robustness to outliers and initially misclassified samples, which single-network boosting-style approaches are sensitive to. These approaches have been primarily explored in a supervised setting. In contrast, we consider an unsupervised setting in which labels are unavailable and, instead, the input signals themselves are corrupted. To the best of our knowledge, such a co-teaching approach has not been explored in the context of denoising.

## 3 PROBLEM SETUP

**Problem Definition.** Given a noisy *target* variable and a reliably measured *auxiliary* time-series, we aim to recover the true values of the target variable. We assume a relationship between the auxiliary

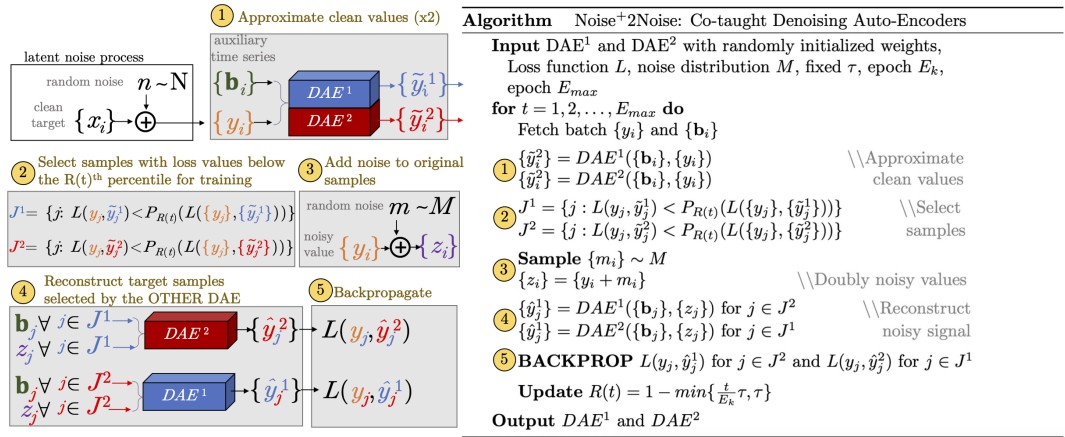

Figure 1: 'Noise$^+$2Noise'. Sample selection is performed with each DAE's output when given the uncorrupted $y$ signal, but backpropagation is performed on the model's output when given a corrupted $y$ signal. The loss values of $DAE^1$ are used to select the sample for backpropagation for $DAE_2$ and vice versa. $P_{R(t)}$ denotes the $R(t)^{th}$ percentile, where $R(t)$ is a function of iteration $t$.

and target variables, and that some samples from the noise distribution associated with the target variable will be close to zero, although which samples is unknown in advance.

**Formalization.** Let $x \in \mathcal{R}$ denote a sample of the target variable. Let $n \sim \mathcal{N}$, where $n \in \mathcal{R}$ denotes a random variable drawn from an unknown distribution $\mathcal{N}$. Let $\mathbf{b} \in \mathcal{R}^T$ denote an auxiliary time-series that is related to $x$, such that a mapping $f(\mathbf{b}) \to x$ exists. Given a sample $y = x + n$ and $\mathbf{b}$, we aim to learn to recover $x$.

**Assumptions.** We assume that the distribution $N$ is independent of both $\mathbf{b}$ and $Y$. We also assume that some values of $n$ are near zero such that given training data with k samples $\{y_i\}_{i=1}^k$ with noise values $\{n_i\}_{i=1}^k$ and true signal values $\{x_i\}_{i=1}^k$, there exists a subset $S$ with sufficient size for training such that the mean and variance of $n_s \forall s \in S$ are negligible compared to the mean and variance of $x_s \forall s \in S$. We assume that the distribution of low noise samples is such that they cover all regions of the input; *i.e.* that $S$ does not exclude entire regions of $Y$. Finally, we assume that the relationship between $\mathbf{b}$ and $x$ can be accurately captured with a recurrent neural network (RNN).

## 4 METHODS

**Overview.** Our method, 'Noise$^+$2Noise' (**N$^+$2N**), is summarized in **Figure 1**. We filter out noisy samples during training, refining the model parameters on selected samples of $y$ estimated to have the least noise. These filtered samples are used to learn to denoise the signal. To identify the samples within a batch to use during training, $y$ and $\mathbf{b}$ are passed to a DAE that outputs $\tilde{y}$, or a de-noised $y$ value. Assuming the DAE accurately reconstructs $x$, the loss $L(\tilde{y}, y)$ represents the expected error between $y$ and $x$. Given a batch $\{y_i\}_{i=1}^n$, we identify the subset of samples with values of $L(\tilde{y}, y)$ below the $R(t)^{th}$ percentile (where $R(t)$ is an increasing function of iteration $t$). These 'low-noise' samples, $y_j \forall j \in J$, are then augmented with additional noise $m_j \sim M$ and, along with corresponding $\mathbf{b}$ vectors, are input to the DAE, which outputs a reconstruction of $y_j$: $\hat{y}_j$. We then backpropagate using the squared error between $\hat{y}_j$ and $y_j$. In this way, the DAE is trained on the samples estimated to have the least noise while utilizing the noisy signal as input without the susceptibility to learning the identity function that comes with standard auto-encoder training. Co-teaching is utilized in that two DAEs are used at each step ($DAE^1$ and $DAE^2$), and during training, the samples identified as low-loss by $DAE^1$ are augmented and passed to $DAE^2$ for backpropagation, and vice-versa. This adds the benefit of ensembling and curtails sensitivity to outliers or incorrectly selected samples. We note that the auxiliary signal $\mathbf{b}$ is input to the model during both the sample selection and backpropagation steps. This allows our approach to function even when the variable of interest $y$ is too noisy or low-dimension to be de-noised alone.

129

**Denoising Autoencoder.** In our setup, additional noise ($m$) is added to $y$ to produce $z$, a 'doubly' noisy measurement of $x$. $z$ and $\mathbf{b}$ are input to a network (henceforth denoted DAE) that outputs $\hat{y} = DAE(z, \mathbf{b})$, and the network is trained to reconstruct $y$: loss is measured between $y$ and $\hat{y}$. As shown by Xu et al. (2020), when the expectation and variance of the noise distribution $N$ are negligible compared to those of the signal, the model parameters that minimize the loss between $\hat{y}$ and $y$ are very close to the optimal parameters of a model trained on clean data. Thus, provided the signal to noise level is high enough, we can pass $y$ to $DAE$ and expect a reduction in noise, with an output much closer to $x$, at inference time.

**Co-teaching DAEs.** We do not expect the noise to be weak in general, but we do assume that some of the samples will be lower noise than others. We identify and train using these samples via an adapted co-teaching approach (Han et al., 2018). We utilize two DAEs, and for each, we backpropagate using only the samples for which the denoised $y$ values from the other DAE are near the original $y$ values. If the denoised $y$ values approach $x$, we are then selecting samples for which the estimated noise $n$ is lowest.

*Claim.* When using co-teaching to train two DAEs ($DAE^1$ and $DAE^2$) in parallel, $\tilde{y}^1 = DAE^1(y, \mathbf{b})$ and $\tilde{y}^2 = DAE^2(y, \mathbf{b})$ approach $x$.

*Justification.* Based on the main result of Xu et al. (2020), these values approximately equal $x$ when a DAE is trained on a dataset where the expectation and variance of the signal are much greater than those of the noise. We have assumed that such a sub-sample exists in our dataset, and we hypothesize that co-teaching will select such a sub-sample. Note that because the noise $n$ is independent of both $y$ and $\mathbf{b}$, any signal learned by $DAE^1$ must be of the form $DAE^1(y, \mathbf{b}) = g(x) + q$, for some function $g$, where $q$ is independent of all variables. The DAE might begin to learn a biased function of $x$; for example, if the noise distribution $N$ is not mean-zero, a DAE might learn the function $g'(x) = x + N$. We curtail this behavior by initializing the DAEs to output the identity function ($y$). Because the model is near the identity function $g(x) = y$ early in training, if it began to learn a biased function such as $g'$, it would begin to output values $g'(x) \approx g'(y) = y + \bar{N}$. These values would be further from the identity than if the model learned a non-biased function of $x$, which would in turn guide the model away from learning $g'$. By similar logic, the model would be encouraged to learn a non-biased function of $x$, the simplest of which is $g(x) = x$. This is only a conjecture, but in practice, we have found that this approach works well even when there is fairly substantial bias in the noise. We note that although the two DAEs are initialized to output the identity, their weights are otherwise random so they are not likely to converge to the same minima. We further prevent convergence by utilizing the co-teaching+ variant (see below).

**Sample selection.** We select the samples with the lowest estimated noise to backpropagate with. If $\tilde{y}^1$ and $\tilde{y}^2$ approach each network's estimated value of $x$, then $DAE^1$'s estimate of $n$, the noise between $x$ and $y$, is approximately $y - \tilde{y}^1$ (and similar for $DAE^2$). For loss function $L$, we use $L(\tilde{y}^1, y)$ and $L(\tilde{y}^2, y)$ to select samples. Given a batch $\{y_i\}_{i=1}^n$, at iteration $t$, we identify the subset of samples with values of $L(\tilde{y}_i^1, y_i)$ below the $R(t)^{th}$ percentile as $J^1 = \{j : L(y_j, \tilde{y}_j^1) < P_{R(t)}(L(\{y_j\}, \{\tilde{y}_j^1\}))\}$, and similarly define $J^2$ for $DAE^2$. As in Han et al. (2018), we begin by training on the full sample. Over the course of training, as the DAEs are expected to become more accurate, we gradually reduce the sample. This prevents the memorization of noisy samples that can occur later in training. Hyper-parameter $\tau \in (0, 1)$ represents the maximum proportion of samples removed and $E_k$ represents the iteration at which we stop increasing the proportion of samples removed. A linear decrease in sample size as a function of iteration $t$ is implemented by using the lowest-loss $R(t) = (1 - Maximum(\frac{t}{E_K}\tau, \tau)) \cdot 100\%$ of samples for backpropagation.

**Training.** Each DAE is trained on the samples for which the other network estimates that the noise is lowest: samples selected by $DAE^1$ ($y_j \forall j \in J^1$) are augmented with additional noise $m_j \sim M$ to generate $z_j$ values. $z_j$, along with corresponding $\mathbf{b}$ vectors, are input to $DAE^2$, which outputs a reconstruction of $y_j$: $\hat{y}_j^2$. We then backpropagate using the squared error between $\hat{y}_j^2$ and $y_j$. Similarly, we only use samples $y_j \forall j \in J^2$, augmented with $m_j \sim M$, to backpropagate $DAE^1$. By selecting samples based on $L(\tilde{y}, y)$ rather than based on $L(\hat{y}, y)$, we are able to select a sample independent of secondary noise value $m$. Selecting samples dependent on $m$ would be confounding because samples might then be selected based on how low the value of $m$ is at the current iteration, rather than the value of $n$, which is hidden. Back-propagation is performed on an input that does not

include clean $y$ values, so the model is not likely to learn the identity function. Sample selection is always performed by the other DAE, so compared to boosting or other one-network approaches, our method is less sensitive to error propagation from wrongly selected samples early in training.

**Co-teaching+.** We utilize co-teaching plus (Yu et al. (2019)), where samples for which the models disagree are selected for backpropagation. As a result, each model learns from the samples for which the other model's estimates were better. This also prevents the models from learning from the samples that they agree upon, which prevents convergence, maintaining unique strengths in each model. We remove the $\sigma\%$ of samples for which the models' outputs are closest (here$\sigma$ is a hyperparameter). This step is performed prior to the sample selection step: the $\sigma\%$ of samples for which the distance between $\hat{y}^1$ and $\hat{y}^2$ are lowest are removed, and then the remaining samples for which $L(\tilde{y}^1, y)$ is lowest are used for the backpropagation of $DAE^2$ and vice versa.

## 5 Real-world Problem Setup: Blood Glucose Management

To explore the benefit of our proposed approach, we consider a real-world problem setup based on blood glucose management that inspired the setting described in Section 3. Nearly two million people in the US have type I diabetes and require insulin to maintain healthy glucose levels due to a pancreatic deficiency in insulin production. Because of this, they must deliver boluses of insulin through an injection or an insulin pump prior to eating to counteract the rise in blood sugar that results from the ingestion of meals. Bolus amounts are calculated based on patient-reported estimates of carbohydrates. Carbohydrates and bolus insulin generally cause blood glucose values to increase or decrease after a delay of 30 minutes to an hour. In our setup, carbohydrates correspond to $x$ values, glucose levels are **b**.

Blood glucose forecasting and control have been extensively studied in the past (Silvia Oviedo, 2016; Fox et al., 2020). Accurate models for blood glucose dynamics are critical to the development of algorithms for managing blood glucose in individuals with diabetes both in terms of patient-selected treatment options and automated solutions. Work in this field is popular largely because the ubiquity of devices for measuring blood glucose and administering insulin make obtaining fairly clean measurements for those values straightforward. However, carbohydrates consumed are patient-reported and as a result are often inaccurate. This in turn leads to inaccurate doses of insulin and poor blood glucose management. Besides misestimation, there are other sources of inconsistency between recorded carbohydrate values and their effects on blood glucose. Variability in meal types is generally poorly captured, which is problematic because the effect of carbs on blood glucose can be moderated by how quickly a meal was consumed, or the amount of protein, fat and other nutrients eaten. Additionally, the timing of a meal may not be recorded accurately. These factors alone make utilizing carbohydrate information difficult, even when carbs are accurately recorded. In an unsupervised setting, denoising approaches could learn representations of carbohydrate values that incorporate these other sources of variability. These representations could be more relevant to blood glucose management than the exact number of grams consumed. This could improve performance of forecast and control algorithms.

## 6 Experimental Setup

We implement our approach in the context of learning to correct noisy patient-reported carbohydrate measurements. We compare performance to several baselines across real and simulated datasets.

### 6.1 Datasets

We utilize two T1D-based datasets. The simulated dataset provides access to ground truth to which we can directly compare our method's denoised outputs. The real dataset provides a more challenging setting for quantifying the efficacy of our approach, but corresponds to real-world scenarios. Both datasets are publicly available and have been previously explored in the context of forecasting and control (Man et al., 2014; Xie, 2018; Marling & Bunescu, 2018; 2020). Both datasets consist of blood glucose, bolus (fast-acting) insulin, basal (slow-acting) insulin, and carbohydrate values. All variables were scaled to be between zero and one. For both datasets, time-series trajectories for each patient were split into windows of 2 hour length ($T = 24$ 5-minute time points). We ignore

windows where a carbohydrate occurs in anywhere but the first position, using only windows with no carbohydrates or carbohydrates at the beginning of the window during training. This means we also ignore windows with more than one carbohydrate present. In a real-world setting these values could be updated recursively, but we simplify our setting here.

**Simulated**. Our primary analyses are performed on data generated with a commonly-used T1D simulator. We use the UVA-Padova simulator (Man et al., 2014) via a publicly available implementation (Xie, 2018). For ten simulated individuals (the "adult" patients modeled in the simulator), we generated approximately 150 days worth of data each, in 30 day roll-outs of the simulator. Days where a patient either had more than 25 timepoints of glucose at the minimum value of 40, or more than 35 timepoints over 450 were thrown out for being non-realistic. The meal schedule used to generate simulated data was based on the Harrison-Benedict equation (Harris & Benedict, 1919) as implemented in (Fox et al., 2020). In our simulation, for all datasets generated, we used the default basal-bolus controller from the existing implementation of the simulator to administer insulin, but we delayed five sixths (randomly selected) of the bolus administrations up to 3.5 hours, with delay time randomly sampled from a uniform distribution. The delay allows for disentanglement between carbohydrate and bolus effects. The carbohydrate values serve as $x$ values, while the CGM values output by the simulator serve as $b$ values. Noise was added to carbs during data generation.

We use noise proportional to the true carb value, as studies on the accuracy of carbe counting report errors relative to the total carbs consumed (Brazeau et al., 2012). Also based on Brazeau et al. (2012), we use a noise distribution with a negative bias, as the carbs were found to be more-often under-reported than not. We therefore set $y = (1 + \mathcal{N}(-.25, .5))x$. We then cap $y$ below and above at 1 and 200 to keep values realistic. We consider additional noise distributions as sensitivity analyses. Bolus values were calculated based on the noisy carbohydrate values. Additionally, 20% of carbohydrates are not reported, to make the dataset more realistic, as missingness is commonplace.

**Real**. This dataset includes both the OHIOT1DM 2018 and 2020 datasets, developed for the Knowledge Discovery in Healthcare Data Blood Glucose Level Predication Challenge (Marling & Bunescu, 2018; 2020). The data pertain to 12 individuals, each with approximately 10,000 5-minute samples for training and 2,500 for testing. 12% of glucose values are missing, but we do not include windows with missing glucose values. We do not include windows with more than one carbohydrate measurement in our analysis. We sum carbohydrates to the first timepoint if they are less than 15 minutes apart to maximize the amount of usable data. We include only individuals with at least 100 training carbohydrate measurements, as fewer than this are not sufficient for learning a model.

## 6.2 BASELINES AND UPPER BOUND

For all non-coteaching methods, we train two DAEs in parallel and report results on their averaged output for a fair comparison. We also note that all models receive the same auxiliary variables (blood glucose/ insulin) as input in an identical fashion.

• **CAE**: An upper performance bound. This model is an autoencoder trained with clean data, which we would expect to perform better than any method without access to clean data.

• **NAC**: Our first baseline. A DAE that treats the noisy data as clean which has been shown to perform well in low noise settings (Xu et al., 2020).

• **NR2N**: Our second baseline is noisier2noise (Moran et al., 2019), which uses the known noise distribution to recover the clean signal. **NR2N** trains similarly to **NAC**, but at evaluation time a transform is used to recover the clean values (briefly, if the distribution of $N$ is known and we set $M = N$, the model should learn to recover half of the noise so the value used at evaluation is $2\hat{y} - z$).
• **SUP**: Our motivating setting can be re-framed as a supervised learning problem: predict $y$ (or $x$) values using **b** values as input. Depending on the noise distribution, it is possible that a model trained on noisy $y$ values could learn to predict the correct $x$, using similar logic to that found in Lehtinen et al. (2018). We therefore use this supervised setting as a naive baseline. We simply input **b** to the same network used in the DAE setting and calculate loss as $(\hat{y} - y)^2$ during training, but here the model has no information regarding $y$ or $z$. As in the DAE setting, at test time we evaluate $(\tilde{y} - x)^2$.
• **SUPCT**: We apply co-teaching to the supervised setting (**SUP**), to ensure that performance gains observed are due to the combination of DAEs and co-teaching, and not co-teaching alone. Here, the model is tuned and trained identically to **N$^+$2N**, except the model does not receive $y$ or $z$ values.

### 6.3 IMPLEMENTATION & TRAINING DETAILS

Each DAE is implemented as a 2-layer bidirectional LSTM with 100 hidden units. The final hidden state is passed to a FC layer with a single output. The output of the model is added to the input value corresponding to $y$, so that the network is tasked with learning the error term rather than a complete reconstruction. This initializes the network's output to be the identity (provided the output is initially low magnitude), which is useful for addressing biased data. Because we only aim to correct a single carbohydrate ($y$) value but use a time-based model, we set one dimension of the model input to be $y$ for all timepoints. Blood glucose values $\mathbf{b}$, bolus and basal insulin are input to the model as time-series. We also carry over bolus insulin values (which occur sparsely) to the end of the input window, to increase their impact on gradient calculations. We threshold the output of each DAE at 0, because carbohydrates (our $x$ and $y$) values cannot be negative. For each co-teaching method and **NR2N**, we tune on a single subject's validation data (adult#001) and use the found parameters for all further analyses. Tuning is described in **Appendix A**. At evaluation we report the result of the average correction learned by both networks when $y$ values are given as input (*i.e.*, where $\tilde{y}_i = DAE_i(y, \mathbf{b})$, we report $L(x, (\tilde{y}^1 + \tilde{y}^2)/2)$).

We split each dataset into training, validation and test sets used for evaluation purposes. For the simulated dataset, we use 80 days for training, 20 for validation, and 50 for testing. For the real dataset, we split the training data into 80% train and 20% validation. The held-out test data were used for evaluation only. We implement and train our models in Pytorch 1.9.1 with CUDA version 10.2, using Ubuntu 16.04.7, a GeForce RTX 2080, an Adam optimizer (Kingma & Ba, 2014), and a batch size of 500. We use a learning rate of 0.01 and a weight decay of $10^{-7}$. We train for at least 500 iterations, and then until validation performance does not improve for 50 iterations, selecting the model for which validation performance was best. For both datasets, we train and test a model on each individual and report across-individual averages. Such individual-specific models/evaluations are common in blood glucose control and forecasting (Silvia Oviedo, 2016), since dynamics vary greatly across individuals and individual-specific training data are typically available

We perform co-teaching on samples containing non-zero $y$ values only. However, when training all models (including baselines) we also pass zero-valued $y$ samples (and their corresponding $\mathbf{b}$ values) through both DAEs and take loss equal to $\hat{y}^2$ for these samples. We do this because there are many more samples with zero-valued carbohydrates than there are with positive values, and this allows the models to learn from this larger corpus. We report results on only positive-valued $y$s, because denoising is only applied to such values.

For sample selection during co-teaching, we use mean squared percentage error $(100\% \cdot ((\hat{y} - y)/y)^2)$ to avoid eliminating all high-valued $y$ samples, as they are likely to have higher noise values. As a noise function during training, we use $z = (1 + \mathcal{N}(0, .5))Bern(.5)y$, *i.e.* we add random noise to half of the samples so that the model can learn to utilize noisy $z$ information, and zero-out the other half so that the model has to learn to distinguish zero from non-zero $y$ values based on $\mathbf{b}$ alone.

### 6.4 EVALUATION

**Simulated Data Metrics.** When clean carbohydrate values are available, we use MSE between the cleaned carbs and true carbohydrate values as our metric to report the remaining noise ($mean((x - \tilde{y})^2)$), where $\tilde{y} = (\tilde{y}^1 + \tilde{y}^2)/2$. The lower this value, the more noise has been removed. Although we assume that these data are not available at training time, we use them for evaluation. Since it can be difficult to interpret the meaning of a difference in MSE, we also consider a clinically motivated evaluation metric: *time in range*. Time in range is a measure of blood glucose management and varies with the accuracy of the carbohydrate measurements. The more accurate the carbohydrate estimates the more time an individual will spend 'in range.' Here, we run a simulation of the subject of interest with the default basal-bolus controller using bolus values calculated from the updated carbohydrate values, and report the proportion of time in the simulation that each individual spent with blood glucose values between 70 and 180, or the euglycemic/healthy range. This metric serves to indicate the real-world impact each approach might have. For both metrics, 95% confidence intervals are calculated for each subject using 1,000 bootstrap re-samples, and the average 2.5 and 97.5 percentiles across subjects are reported.

**Sensitivity analyses.** To evaluate our model under different noise assumptions, we repeat our analysis with multiple noise generation methods ($x \rightarrow y$), without altering hyperparameters or our $y \rightarrow z$

noise function. We use various Gaussian and uniform distributions reported in **Appendix B**, which include zero and negative mean multiplicative and additive noise functions. We do not aim at a comprehensive evaluation of all possible noise types, but rather we aim to include various distributions that are likely similar to those that might arise in our motivating domain.

**Real Data analysis.** Without access to clean carbohydrate values at test time for the real dataset (unlike the simulated dataset), we evaluate the performance of our denoising approach based on a proxy. We take advantage of the fact that poorly estimated carbohydrates result in inaccurate bolus calculations, which result in poor blood glucose management. We expect that inaccurate carbohydrates estimates (large $[x - y]$ values) result in the poor blood glucose management. For a model that has come close to estimating $x$ correctly, we would observe a correlation between $[\tilde{y} - y]$ values and blood glucose control in the time period following a meal. We assess this with Correction-Risk-Correlation, henceforth referred to as CRC, defined as the Spearman correlation between the squared carbohydrate correction value $((\tilde{y} - y)^2)$ and the average Magni Risk (Magni et al., 2007) of blood glucose in the second hour following the carbohydrate. We use the Spearman correlation to account for non-linearities in the risk and correction value distributions. Magni risk is a measure of how far from a safe value blood glucose is; higher risk values correspond to blood glucose values that are either dangerously high or dangerously low. We use the second hour following the carbohydrate because the effects of the carbohydrate consumption and insulin bolus have not fully taken effect in the first hour. We calculate this correlation across all carbohydrates observed in all individuals. For validation purposes, we also calculate this metric for the simulated dataset.

# 7 RESULTS AND DISCUSSION

Through our experiments, we aim to answer the following questions. 1) Does our approach meaningfully reduce error across a variety of simulated individuals, compared to existing approaches? 2) Is our model robust to different domain-appropriate noise distributions? 3) Does our model show strong performance in real data, indicating accurate denoising?

Table 1: Our approach outperforms the baselines with respect to all of the evaluation metrics and across both SIM and REAL datasets, falling only 2% short of the upperbound w.r.t. the clinical measurement of 'Time in Range.' 95% CIs are calculated from 1,000 bootstrap re-samples.

| Model | Remaining Carb MSE ($g^2$,SIM) | % Time in Range (SIM) | SIM CRC (r,p) | REAL CRC (r,p) |
|---|---|---|---|---|
| N/A-clean carb | 0.00 (0.00, 0.00) | 73.18 (72.49,73.88) | N/A | N/A |
| N/A-noisy carb | 72.26 (54.16, 92.58) | 65.43 (64.70,66.16) | N/A | N/A |
| CAE | 6.96 (5.03, 9.18) | 72.44 (71.76,73.11) | 0.32, 4e-31 | N/A |
| | | | | |
| SUP | 58.37 (45.11, 73.31) | 64.59 (63.89,65.30) | 0.04, 0.14 | 0.19, 3e-3 |
| SUPCT | 100.50 (84.12,118.40) | 60.22 (59.43,60.94) | 3e-3, 0.91 | 0.03, 0.61 |
| NAC | 36.40 (27.02, 46.95) | 68.22 (67.53,68.93) | 0.11, 1e-4 | 0.13, 0.05 |
| NR2N | 33.44 (26.59, 43.23) | 68.79 (68.09,69.51) | 0.11, 1e-4 | 0.13, 0.05 |
| N$^+$2N (Ours) | **17.91 (12.61, 24.98)** | **71.24 (70.54,71.90)** | **0.19, 3e-11** | **0.22, 9e-4** |

**Error Reduction for simulated data.** Co-taught denoising autoencoders, **N$^+$2N**, outperform baselines in terms of noise reduction (MSE) (**Table** 1). **N$^+$2N** reduces MSE to $18g^2$, but fall short of the value achieved by the clean-data DAE ($7g^2$), as expected. **CAE** does not achieve perfect MSE, probably due to insufficient training data or to a small amount of noise in the CGM signal. Our approach's reduction in noise is meaningful since it leads to significantly better time in range. Most methods offer an improvement in % time in range when used in a basal bolus controller, with baselines increasing over the noisy value from 65% to 69%, and **N$^+$2N** further improving performance to 71%, recovering 6% time in range out of a total of 8% lost when using noisy versus clean values. **SUPCT** performs worse than any other method including **SUP**, likely because without the noisy carbohydrate measurement as input, co-teaching cannot learn the relationship between **b** and $y$ as easily, and therefore does not identify the less corrupted samples during training. This results in essentially random sub-sample selection, hampering performance as less training data becomes available. **SUP** does not suffer from this problem because it utilizes the entire dataset for all iterations.

**Alternate noise assumptions.** **N$^+$2N** outperforms all baselines across the majority of noise distributions (**Figure** 2). For zero-mean uniform multiplicative noise, **NAC** outperforms the proposed

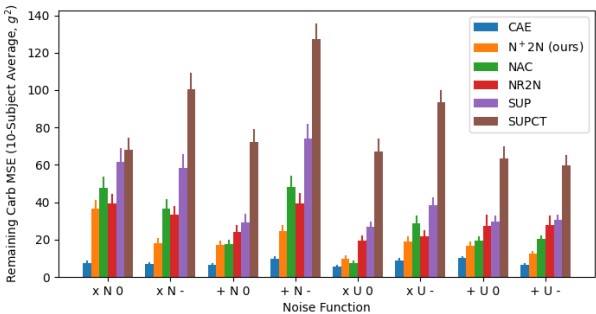

Figure 2: Performance on datasets with multiplicative ($\times$) vs. additive ($+$), Normal ($N$) vs. Uniform ($U$), and zero (0) vs. negative mean ($-$) noise functions. $\mathbf{N^+2N}$ generally outperforms baselines.

approach. We hypothesize that **NAC** performs well in this setting because the expected value of the noise is zero and the variance is lower than in other settings (it is 33%, which is approximately 30g, compared to 75% in the multiplicative normal setting, or 40g and 60g in the additive noise settings, see **Appendix B**), which is the setting for which **NAC** has been shown to perform best. Of note, this analysis was carried out without additional tuning, demonstrating the resilience of our approach to varying noise assumptions. Across biased noised distributions, our proposed approach consistently outperforms *all* baselines. This resilience is likely due to our approach's identity-initialized sample selection and a lack of dependence on matching $M$ and $N$ noise distributions.

**Experiments on Real Data.** For the real dataset, $\mathbf{N^+2N}$ outperforms all baselines with respect to CRC. For simulated data, we see that, without exception, models with lower remaining MSE after denoising have a higher or equal CRC. This indicates that our metric serves as a reasonable proxy for remaining error when true values are unavailable. Plots showing the components used to calculate CRC (magnitude of carbohydrate correction versus Magni risk an hour after the meal) can be found in **Appendix C**. Interestingly, the **SUP** baseline performs fairly well for this task on the real dataset. We hypothesize that this may be because carbohydrate measurements are so unreliable for this dataset that learning to predict them from scratch (without access to noisy values at test time) is sufficient. Four out of twelve individuals in the real dataset had too few carbohydrate measurements to be included in our analysis, indicating that more work would need to be done for our algorithm to be applicable to a broader population.

## 8 CONCLUSIONS

We propose a new approach to denoising, 'Noise$^+$2Noise', that does not assume access to clean samples and applies to time-series data. Our approach leverages an auxiliary time-series that is related to the target signal to help identify target samples with less noise. Our approach is the first to adapt co-teaching to de-noising. In the context of carbohydrate recovery for blood glucose management, compared to existing approaches, 'Noise$^+$2Noise' leads to better signal reconstruction that is both statistically significant and clinically significant. While promising, our approach is not without limitations. Our primary analyses are on simulated data where ground truth labels are available, but in real datasets common evaluation metrics (e.g., MSE) do not apply and we must rely on proxies. As presented, our approach is designed for retrospective carbohydrate correction; more work is necessary to investigate its applicability to closer-to-real-time correction. While our approach shows clear significance for the individuals available, evaluation on a much larger population would be necessary prior to widespread adoption. Finally, while we have empirically shown that co-teaching appears to select a low-noise sample, we have not proven any statistical guarantees. Despite these limitations, we have demonstrated that is is feasible to correct a noisy variable without access to clean samples, expanding the utility of ideas the image analysis and noisy label learning to time-series reconstruction. Applied to domains in which datastreams are composed of both individual-reported data and data measured from relioable sensors (e.g., mHealth), or datastreams composed of series with disparate noise levels (e.g., external vs internal temperature, multiple audio recordings), our approach could aid in improving the reliability of uncertain time-series data.

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

## 9 APPENDICES (SUPPLEMENTAL)

## A TUNING DETAILS

For each model, tuning was performed on simulated adult#001 using validation performance. No additional tuning was performed for other individuals or noise functions. For Noisier2Noise, we selected $\alpha$, the parameter that controls the relative noise distributions, from [0.1,0.3,0.5,0.7,0.9,1.0,1.25,1.5,1.75,2], ultimately selecting $\alpha = 1$. Because we do not assume access to the exact noise we would not expect this method to perform spectacularly, but note that it often outperforms other baselines.

For co-teaching methods, we performed a simple grid search over the values of $E_k$=[250,500] (where 500 is the minimum number of training iterations), $\tau$=[0.333,0.5,0.667], and $\sigma =$ [0.1,0.3,0.5,0.7]. For **N$^+$2N**, we selected $E_k = 250$, $\tau = 0.333$, and $\sigma = 0.1$. For the supervised setting co-teaching (**SUPCT**), we set selected $T_k = 500$, $\tau = 0.5$ and $\sigma = 0.3$.

## B ALTERNATE NOISE FUNCTIONS

We consider noise functions that might arise in carbohydrate counting. None are highly dissimilar from our main analysis noise function: we aim here at feasibility, rather than a comprehensive survey on a broad selection of loss functions, which our method would likely be unable to address without further tuning or modification. Here, $\mathcal{U}(a, b)$ denotes a uniform distribution with values between $a$ and $b$. Carbohydrate values range between 0 and 200. After adding noise, $y$ values are capped above and below by 1 and 200. Alternate noise functions include:

1. Zero-mean multiplicative Gaussian: $y = (1 + \mathcal{N}(0, .75))x$
2. Negative-mean multiplicative Gaussian (primary noise function): $y = (1 + \mathcal{N}(-.25, .5))x$
3. Zero-mean additive Gaussian: $y = x + \mathcal{N}(0, 40)$
4. Negative-mean additive Gaussian: $y = x + \mathcal{N}(-30, 50)$
5. Zero-mean multiplicative Uniform: $y = \mathcal{U}(.5, 1.5)x$
6. Negative-mean multiplicative Uniform: $y = \mathcal{U}(0, 1.6)x$
7. Zero-mean additive Uniform: $y = x + \mathcal{U}(-60, 60)$
8. Negative-mean additive Uniform: $y = x + \mathcal{U}(-60, 40)$

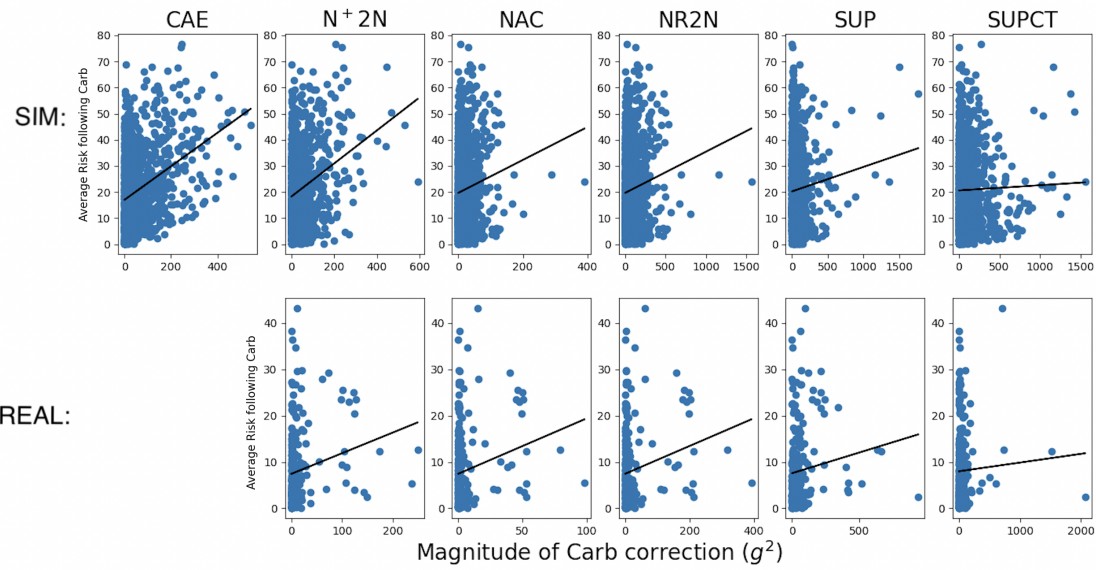

Figure 3: Risk following the carbohydrate vs. magnitude of carbohydrate correction learned for all models and both datasets. Besides the clean autoencoder, $N^+2N$ performs best.

## C   CRC PLOTS

With $N^+2N$, we see a higher correlation between the magnitude of carbohydrate correction and risk following the meal compared to baselines for both real and simulated data (**Figure 3**).

