# OpenReview forum: "Noise$^+$2Noise: Co-taught De-noising Autoencoders for Time-Series Data"
_ICLR.cc/2023/Conference — Submitted to ICLR 2023_

### Official Review · Reviewer_Azxw · 2022-10-26

**Confidence:** 3
**Correctness:** 3
**Technical Novelty And Significance:** 2
**Empirical Novelty And Significance:** 3
**Recommendation:** 6

**Clarity, Quality, Novelty And Reproducibility:**

The paper is written clearly, but I would wish the method was laid out more clearly.

**Strength And Weaknesses:**

STRENGTHS:
1. This works is inspired by a real-world problem and contributes a method that leads to very good results on that problem.
2. The authors are evaluating the model robustness wrt. different noise distributions and are showing that their work outperforms all baselines for most data.

WEAKNESSES:
1. I would have appreciated if the authors would have elaborated more on the usage of the auxiliary noise-free time series in the structure of the network (in both steps).
2. While useful, the proposed method lacks novelty.
3. It is not clear to me what the reason are for choosing any specific threshold R(t) (for L).


**Summary Of The Paper:**

This manuscript addresses the problem of recovering a clean signal given only the noisy data using denoising autoencoders in the context of time series data. Taking advantage of time series data, the authors use an auxiliary noise-free time series signal that is related to the target signal and a co-teaching-based approach for training DAEs. The proposed model can leverage the relationship between the auxiliary and target signals and does not get affected by not having a weak SNR. They show the benefit of their approach by using a real-world problem setup, i.e. blood glucose forecasting and control.

**Summary Of The Review:**

A nice paper that addresses a relevant task, builds upon existing ideas. More elaboration could be given to the details of the method itself.

---

> ### Author Response · Authors · 2022-11-16
> **Response to Azxw**
>
> We thank the reviewer for their valuable feedback and suggestions. We are encouraged that the reviewer appreciated our approach’s applicability to a real-world setting and demonstrated robustness. We address the weaknesses identified below:
>
> 1. Usage of auxiliary noise-free signal
>
> We have updated the methods sections and the model figure to better elaborate on the usage of the secondary signal: it is simply input to the auto-encoder which allows it to be utilized implicitly, at both steps.
>
> Page 3 Line 126:
>
> “We note that the auxiliary signal $\textbf{b}$ is input to the model during both the sample selection and backpropagation steps. This allows our approach to function even when the variable of interest $y$ is too noisy or low-dimension to be de-noised alone.”
>
> 2. Novelty
>
> While a similar co-teaching method has been proposed in the context of noisy label learning, such a method has never been applied to de-noising auto-encoders, the de-noising of variables generally, or to continuous metrics. We made significant modifications to the method to allow it to be applicable to our setting, and believe that showing its applicability to a completely different task is novel and valuable in its own right. We also note that, to date, no deep method has been proposed that can address noisy-data only time series de-noising. We have clarified the novelty of the approach in the revised introduction and conclusion sections:
>
> Page 2 Line 47:
>
> “This co-teaching approach has never been applied to de-noising in a time-series or any other setting. It has also never been utilized in a continuous output setting. By adapting this co-teaching approach to DAEs, we provide a solution to denoising in this novel (time-series) setting.”
>
>
> Page 9 Line 403:
>
>  ”Our approach is the first to adapt co-teaching to de-noising.”
>
> 3. R(t) threshold
>
> R(t) is calculated so that training starts out utilizing the full sample, and then the sample size is linearly decreased at each iteration until it reaches some final proportion of the original sample (this final proportion is a tuned hyperparameter). The purpose of this is to select a more refined low-noise sample as the model becomes more accurate, which both avoids over-fitting and allows the model to be trained on an optimally clean sample (ideally). Details about this have been added to the current draft’s method section.
>
> Page 4 Line 173:
>
> “A linear decrease in sample size as a function of iteration $t$ is implemented by using the lowest-loss $R(t)=(1-Maximum(\frac{t}{E_K}\tau,\tau))\cdot 100\%$ of samples for backpropagation.”
>
> 4. Clarity of methods section.
>
> We have worked to clarify the methods section by significantly updating Figure 1 to better illustrate co-teaching and cross-DAE sample selection and to include pseudo-code (Page 3).  We thank the reviewer for noting this area for improvement. We also edited the methods section, including various edits to enhance syntactical clarity as well as:
>
> Page 3 Line 126, we clarify the use of the auxiliary signal (see point 1)
>
> Page 4 line 163, we begin the section with a clearer summary:
>
> “We select the samples with the lowest estimated noise to backpropagate with.”
>
> Page 4 line 173, we better describe the calculation of R(T) (see point 3)
>
> Page 4 line 175, we begin the section with a clearer summary:
>
> “Each DAE is trained on the samples for which the other network estimates that the noise is lowest”

---

### Official Review · Reviewer_reMC · 2022-10-29

**Confidence:** 4
**Correctness:** 2
**Technical Novelty And Significance:** 3
**Empirical Novelty And Significance:** 3
**Recommendation:** 6

**Clarity, Quality, Novelty And Reproducibility:**

In general Figure 1 is not clear. While it can be understood somewhat after reading the text, overall it is a missed opportunity to give the procedure the co-teaching scheme graphically. Instead, it mostly serves as a separate textbox. Second, the presentation of the same content of the text is dense. This is understandable given careful reading, but it is the most dense portion of the text where it should be the most lucid.

Overall, this seems like a novel application of co-teaching, and seems simple enough to reproduce with the explicit exception dependence on the time-series/side-channel data.

**Strength And Weaknesses:**

Strengths:
* The overall co-teaching DAE idea is novel and interesting, in my opinion.
* Experimental results are fully fleshed out, in that the baselines and upper bounds show many of the experimental conditions/ablations that I desired to see, e.g. a supervised case, a clean data upper bound, etc..

Weaknesses:
* My major theoretical concern are cases where the hardness of differing samples under (stationary) Gaussian noise is not uniform. If the data domain is biased toward easy denoising in specific regions and hard problems in others, then it seems that this dual DAE scheme will not reconstruct the relatively harder regions where both DAEs fail, excluding them from the training set. While it is reasonable to ascribe this high error to high noise, it is also reasonable to ascribe this to the x domain also; this confounding seems unavoidable with the presented co-teaching scheme.
* The entire framing of the timeseries data (in the title, section 1, etc.) seems superfluous, except for the fact that the timeseries are used as side channel data. Is the additional empirical performance due to the fact that the proposed method is the only noise2noise method that includes the side-data? I think applying the NAC baseline to an expected y output would be a good experiment to test this proposition. It's not clear to me either way, but the entire side-channel/time-series context makes the rest of the analysis more complicated in my opinion.

**Summary Of The Paper:**

The authors propose an unsupervised denoising method for situations in the high noise regime with side information. They frame the problem for time-series applications, though the proposed method appears general. Their method is a variant of the Noise2Noise method (Lehtinen et al 2018), which itself builds off of Denoising Auto-encoders (Vincent et al 2008). They propose using dual independently initialized auto encoders to find subsets of the training set with low error on one encoder but not the other, assigning the remaining , which mark "easy" or nearly correct examples. They then learn using only these samples.

They present computational results on a

**Summary Of The Review:**

A novel dual DAE scheme involving co-teaching, with strong empirical results, but strange choices with respect to side data, and one particular missing experiment.

---

> ### Author Response · Authors · 2022-11-16
> **Response to reMC**
>
> We thank the reviewer for their valuable feedback and suggestions. We are encouraged that the reviewer found our main idea to be interesting and thought that our experiments were well fleshed-out. We address the weaknesses identified below:
>
> 1. Non-uniform noise
>
> This is an excellent point, and we thank the reviewer for identifying this implicit assumption that we failed to make explicit. Our method assumes that the distribution of noise, while not necessarily uniform, must have low SNR samples across all regions of the input so that the model does not select an overly biased or censored training set. This has been added to the assumptions section.
>
> Page 3 Line 107:
>
> “We assume that the distribution of low noise samples is such that they cover all regions of the input; \textit{i.e.} that $S$ does not exclude entire regions of $Y$.”
>
> 2. Time series framing and auxiliary data usage
>
> We do not know of any other deep method that is able to address this setting (noisy-data only denoising in time-series). We note that the side channel and variable of interest are both time-series, and our model is built entirely on an RNN and is therefore only applicable to time-series. The side channel is absolutely necessary in this setting; the carbohydrate input is univariate and sparse, meaning that any reasonable-length model input would only have a single value, which would be impossible to denoise. All baseline methods include the side-data; they all utilize the same LSTM architecture which takes the auxiliary signal as input in an identical way. Our proposed approach does not make any additional use of the auxiliary input. This has been made clearer in the methods and experimental setup sections of the revised manuscript.
>
> Page 3 Line 126:
>
> “We note that the auxiliary signal $\textbf{b}$ is input to the model during both the sample selection and backpropagation steps. This allows our approach to function even when the variable of interest $y$ is too noisy or low-dimension to be de-noised alone.”
>
> Page 6 Line 268:
>
> “We also note that all models receive the same auxiliary variables (blood glucose/ insulin) as input in an identical fashion.”
>
>
>
> 3. Clarity of figure 1 and methods section
>
> We have worked to clarify the methods section by significantly updating Figure 1 to better illustrate co-teaching and cross-DAE sample selection and to include pseudo-code (Page 3).  We thank the reviewer for noting this area for improvement. We also edited the methods section, including various edits to enhance syntactical clarity as well as:
>
>
> Page 3 Line 126, we clarify the use of the auxiliary signal (see point 2).
>
> Page 4 line 163, we begin the section with a clearer summary:
>
> “We select the samples with the lowest estimated noise to backpropagate with.”
>
> Page 4 line 173, we better describe the calculation of R(T):
>
> “A linear decrease in sample size as a function of iteration $t$ is implemented by using the lowest-loss $R(t)=(1-Maximum(\frac{t}{E_K}\tau,\tau))\cdot 100\%$ of samples for backpropagation.”
>
> Page 4 line 175, we begin the section with a clearer summary:
>
> “Each DAE is trained on the samples for which the other network estimates that the noise is lowest”

---

### Official Review · Reviewer_SnFL · 2022-10-29

**Confidence:** 3
**Correctness:** 4
**Technical Novelty And Significance:** 2
**Empirical Novelty And Significance:** 2
**Recommendation:** 5

**Clarity, Quality, Novelty And Reproducibility:**

Clarity

The paper is sufficiently clear, although the complexity the algorithm makes it a bit difficult to easily understand parts of the methods section.

Quality

The paper's approach seems sound and it is well validated. The novelty and the applicability of the algorithm to real-world problems is limited as described in the weaknesses of the paper section above, which reduces the quality of this submission.

Novelty

Very limited, authors leverage already existing approaches and apply them to time series data. The novel result is that those already known approaches also work for time-series data.

Reproducibility

The authors provide enough information to enable others to reproduce their work. I have not personally attempted to reproduce the key results of the paper, but I believe this will be easily possible once the source code is released.

**Strength And Weaknesses:**

Strengths

The paper's approach is well validated (using synthetic data) and clearly compared to relevant related methods. The performance of the algorithm is shown to be on par or superior to that of comparable algorithms.

Weaknesses

The method itself is not really novel. As the authors state in their paper, their work mostly shows that approaches which had already been shown to work with image data can also be applied to time series data. The real-world application of the paper, although compelling, is also very limited and not really feasible. As the authors themselves explain in the paper, their approach only works to de-noise data retrospectively and cannot work for real-time de-noising as presented in the paper.
The method itself also makes a number of strong assumptions that further limits its potential applicability. It assumes that there exists a clean signal that is related to the target signal, and that the co-teaching method can be sufficient to increase the SNR enough for the algorithm to recover a clean version of the target.

**Summary Of The Paper:**

Authors present a de-noising method using auto-encoders for time series data that does not need clean data to be trained. However, the method requires an auxiliary clean signal that is related to the target signal. The method also makes assumptions about the SNR level which has to be high. To alleviate this limitation, the authors adapt a co-teaching technique to choose the less noisy examples from the training data. The authors validate their approach using synthetic data and demonstrate that the performance of their algorithm is on par or superior to similar methods. Authors illustrate the applicability of their proposal by tackling a real-world problem blood-glucose management.

**Summary Of The Review:**

This submission is interesting and it is well executed, however the novelty and applicability are very limited and therefore its impact is small. For these reasons, I think that this submission is a bit below the threshold of the conference.

---

> ### Author Response · Authors · 2022-11-16
> **Response to SnFL**
>
> We thank the reviewer for their valuable feedback and suggestions. We are encouraged that the reviewer found our approach to be well validated. We address the weaknesses identified below:
>
> 1. Novelty vs image techniques
>
> While aspects of our approach were inspired by work in computer vision, our approach incorporates and modifies co-teaching, which has never been applied to de-noising images or to any other de-noising setting.  We propose a novel approach to address the task of reconstructing time series without clean measurements. The combination of co-teaching and denoising auto-encoders, and applying co-teaching to a noise-recovery setting generally, are also unique. We clarify the novelty in the introduction and conclusions section of our updated draft:
>
> Page 2 Line 47:
>
> “This co-teaching approach has never been applied to de-noising in a time-series or any other setting. It has also never been utilized in a continuous output setting. By adapting this co-teaching approach to DAEs, we provide a solution to denoising in this novel (time-series) setting.”
>
> Page 9 Line 403:
>
>  “Our approach is the first to adapt co-teaching to de-noising.”
>
>
> 2. Applicabilicability of retrospective correction
>
> We did not intend to present our results as real-time denoising and have worked to clarify this up front in the revised introduction, and also list it as a limitation in the revised conclusion.
>
> Page 2 Line 58:
>
> “Our approach aims to retrospectively correct measurements of patient-reported signals, like carbohydrates in the context of diabetes, which could lead to improved treatment decisions: patients can learn when they are over- or under-reporting and adjust in the future, ultimately improving health outcomes.”
>
> Page 9 Line 408:
>
> “As presented, our approach is designed for retrospective carbohydrate correction; more work is necessary to investigate its applicability to closer-to-real-time correction.”
>
>
>
> 3. Assuming access to a clean auxiliary variable and the efficacy of co-teaching for sample selection
>
> Beyond managing blood glucose, we believe that there are many other settings where the assumption of access to a clean auxiliary variable is reasonable. Any setting in which there is a time-series forecasting task based on inter-dependent variables with disparate noise levels (e.g., settings with subjective and objective measurements or measurements of varying reliability) could be relevant. We have updated the discussion accordingly to acknowledge this.
>
> Page 9 Line 416:
>
> “Applied to domains in which datastreams are composed of both individual-reported data and data measured from wearable sensors (e.g., mHealth), or datastreams composed of series with disparate noise levels (e.g., external vs internal temperature, multiple audio recordings),  our approach could aid in improving the reliability of uncertain time-series data.”
>
>
> We believe that the co-teaching method is able to select a sample with a higher SNR based on our empirical results. We do not think that it is unreasonable to assume that some low noise samples exist within the sample, especially since much real world noise is mean zero. We note this assumption is made explicit in the text (page 3 line 103). While we do not have a proof that co-teaching will select a low noise sample, we note that this claim is consistent with our results and the results of papers that have examined co-teaching in the noisy-label learning setting. We also provide a justification for the assumption (page 4 line 146). However, we do note that this is a limitation and add it to the revised conclusions section:
>
> Page 9 Line 412:
>
> “Finally, while we have empirically shown that co-teaching appears to select a low-noise sample, we have not proven any statistical guarantees.”
>
>
>
> 4. Clarity of Methods Section
>
> We have worked to clarify the methods section by significantly updating Figure 1 to better illustrate co-teaching and cross-DAE sample selection and to include pseudo-code (Page 3). Other changes include various small edits to enhance syntactical clarity as well as:
>
> Page 3 Line 126, we clarify the use of the auxiliary signal:
>
> “We note that the auxiliary signal $\textbf{b}$ is input to the model during both the sample selection and backpropagation steps. This allows our approach to function even when the variable of interest $y$ is too noisy or low-dimension to be de-noised alone.”
>
> Page 4 line 163, we begin the section with a clearer summary:
>
> “We select the samples with the lowest estimated noise to backpropagate with.”
>
> Page 4 line 173, we better describe the calculation of R(T):
>
> “A linear decrease in sample size as a function of iteration $t$ is implemented by using the lowest-loss $R(t)=(1-Maximum(\frac{t}{E_K}\tau,\tau))\cdot 100\%$ of samples for backpropagation.”
>
> Page 4 line 175, we begin the section with a clearer summary:
>
> “Each DAE is trained on the samples for which the other network estimates that the noise is lowest.”

---

### Official Review · Reviewer_yPFM · 2022-11-04

**Confidence:** 3
**Correctness:** 2
**Technical Novelty And Significance:** 3
**Empirical Novelty And Significance:** 2
**Recommendation:** 3

**Clarity, Quality, Novelty And Reproducibility:**

* The paper is present clearly and easy to understand.
* The DAE model for time-series data seems novel. But the co-training method has been proposed before.
* Reproducibility - not sure.

**Details Of Ethics Concerns:**

The way the proposed method is validated seems to be incorrect. The conclusions from this paper may not be reliable, which poses risks for real medical applications.

**Strength And Weaknesses:**

Strength:
* The paper is clearly written and the method is carefully described
* The problem setup and approach are novel

Weaknesses:
* The experimental setup is flawed, where the authors split train/valid/test based on time for each individual. This means that individuals presented in the training set are identical to the test set. This is unrealistic and cannot demonstrate generalization of the model.
* Although collecting data for this task is challenging, only 10 individuals in the simulated dataset and 12 individuals in the real dataset are just too few.
* Applications of the proposed method is very limited.

**Summary Of The Paper:**

This paper proposes to recover clean data from corrupted inputs by denoising audoencoders (DAEs) in an unsupervised setup. Specifically it assumes a situation where ground truth does not exist but related noisy targets and a clean auxiliary signal exist. Authors proposed a co-training approach for DAE models to leverage the noisy target that is related to the clean target. This approach was applied to the task of recovering carbohydrate values for blood glucose management, where the carbohydrate values are the noisy data and the blood glucose values are the clean auxiliary signal.

**Summary Of The Review:**

Although the proposed method seems novel and interesting for the particular problem setup, the authors' experimental setup is flawed and the potential applications are quite limited.

---

> ### Author Response · Authors · 2022-11-16
> **Response to yPFM**
>
> We thank the reviewer for their valuable feedback and suggestions. We are encouraged that the reviewer found our paper to be clearly written and the approach to be novel. We address the weaknesses identified below:
>
>
> 1. Data-split based on time for each individual.
>
> We appreciate the reviewer’s concern here, as we understand that a contaminated testing set would invalidate our results, and in many domains a within-subject split would be unacceptable. However, in blood glucose control and forecasting and other tasks associated with managing type 1 diabetes, models are nearly always patient-specific i.e., trained and tested for a single patient at a time (see A review of personalized blood glucose prediction strategies for T1DM patients, Int J Numer Method Biomed Eng 2017). We clarify in the revised experimental setup section.
>
> Page 7 Line 309:
>
> “For both datasets, we train and test a model on each individual and report across-individual averages. Such individual-specific models/evaluations are common in blood glucose control and forecasting (Oviedo 2017), since dynamics vary greatly across individuals and individual-specific training data are typically available.”
>
> 2. Number of individuals
>
> We appreciate that in many domains, especially where some individuals must be held out as a test set, 18 individuals would be too few for a meaningful evaluation. While this might be the case in other domains, given the complexity around using real patient data, evaluating on this number of individuals is common practice in blood glucose management for diabetes. The Ohio dataset used in our experiments is a benchmark that has been used for blood glucose forecasting for years, and similar sized study cohorts are common (see Deep Learning for Diabetes: A Systematic Review, IEEE JOURNAL OF BIOMEDICAL AND HEALTH INFORMATICS 2021). We believe that the clear statistical significance of our results across individuals is sufficient to prove generalizability. That said, we now acknowledge the clear limitations of the evaluation in the updated conclusions section.
>
> Page 9, Line 410:
>
> “While our approach shows clear significance for the individuals available, evaluation on a much larger population would be necessary prior to widespread adoption.”
>
> 3. Potential Applications
>
> Managing blood glucose is a significant problem in healthcare. While we focus on individuals with T1D, this work could extend to individuals with type 2 diabetes, which is nearly 500 million individuals globally. We chose this domain because of its broad applicability and our availability of data and domain expertise (we work closely with clinical practitioners in T1D).  Beyond managing blood glucose, we believe the approach could apply more to other settings in which there are forecasting tasks based on inter-dependent variables with disparate noise levels (e.g., settings with subjective and objective measurements or measurements of varying reliability). We have updated the discussion accordingly.
>
> Page 9 Line 416:
>
> “Applied to domains in which datastreams are composed of both individual-reported data and data measured from reliable sensors (e.g., mHealth), or datastreams composed of series with disparate noise levels (e.g., external vs internal temperature, multiple audio recordings),  our approach could aid in improving the reliability of uncertain time-series data.”
>
> 4. Novelty of Co-teaching
>
> While a similar co-teaching method has been proposed in the context of noisy label learning, its use in a denoising autoencoder is novel. For co-teaching to work in this continuous output setting, we required some clever adaptations. In addition, we believe that demonstrating its applicability to a completely different task is a valuable contribution in its own right. We emphasize our contributions  in the introduction and conclusion of the updated manuscript.
>
> Page 2 Line 47:
>
> “This co-teaching approach has never been applied to de-noising in a time-series or any other setting. It has also never been utilized in a continuous output setting.”
>
>
> Page 9 Line 403:
>
>  “Our approach is the first to adapt co-teaching to de-noising.”

---

### Decision · Program_Chairs · 2023-01-20

**Decision:**

Reject

**Justification For Why Not Higher Score:**

I believe ICLR papers should have significant development in the understanding of methodology. The current paper is lacking in the "understanding" part and is not interesting to the general ICLR audience.

**Justification For Why Not Lower Score:**

N/A

**Metareview: Summary, Strengths And Weaknesses:**

This paper proposes to recover clean data from corrupted inputs by denoising audoencoders (DAEs) in an unsupervised setup. Specifically it assumes a situation where ground truth does not exist but related noisy targets and a clean auxiliary signal exist. Authors proposed a co-training approach for DAE models to leverage the noisy target that is related to the clean target. This approach was applied to the task of recovering carbohydrate values for blood glucose management.

Strength:

The paper is clearly written and the method is carefully described. The problem setup and approach are interesting.

Weaknesses:

The experimental setup is quite different from that of typical machine learning applications, it is hard to tell if the methodology is generalizable to more typical/popular domains (audio, video). The method is not backed up by rigorous/theoretical arguments; the reasoning in Section 4 is not formal and appears hand-wavy. Due to the lack of technical development and evidence of wide applicability, I think the paper is more suitable for a specialized venue than ICLR.